# Transcriptomic Insight into the Pollen Tube Growth of *Olea europaea* L. subsp. *europaea* Reveals Reprogramming and Pollen-Specific Genes Including New Transcription Factors

**DOI:** 10.3390/plants12162894

**Published:** 2023-08-08

**Authors:** Amanda Bullones, Antonio Jesús Castro, Elena Lima-Cabello, Noe Fernandez-Pozo, Rocío Bautista, Juan de Dios Alché, Manuel Gonzalo Claros

**Affiliations:** 1Department of Molecular Biology and Biochemistry, Universidad de Málaga, 29010 Malaga, Spain; amandabullones@uma.es; 2Institute for Mediterranean and Subtropical Horticulture “La Mayora” (IHSM-UMA-CSIC), 29010 Malaga, Spain; noe.fernandez.pozo@csic.es; 3Plant Reproductive Biology and Advanced Imaging Laboratory (BReMAP), Estación Experimental del Zaidín (EEZ-CSIC), 18008 Granada, Spain; antoniojesus.castro@eez.csic.es (A.J.C.); elena.lima@eez.csic.es (E.L.-C.); juandedios.alche@eez.csic.es (J.d.D.A.); 4Plataforma Andaluza de Bioinformática, Supercomputing and Bioinnovation Center (SCBI), Universidad de Málaga, 29590 Malaga, Spain; rociobm@uma.es; 5University Institute of Research on Olive Grove and Olive Oils (INUO), Universidad de Jaén, 23071 Jaen, Spain; 6CIBER de Enfermedades Raras (CIBERER) U741, 29071 Malaga, Spain; 7Institute of Biomedical Research in Málaga (IBIMA), IBIMA-RARE, 29010 Malaga, Spain

**Keywords:** germination, olive, pollen, pollen tube, RNA-seq, transcriptomics, bioinformatics, reprogramming

## Abstract

The pollen tube is a key innovation of land plants that is essential for successful fertilisation. Its development and growth have been profusely studied in model organisms, but in spite of the economic impact of olive trees, little is known regarding the genome-wide events underlying pollen hydration and growth in this species. To fill this gap, triplicate mRNA samples at 0, 1, 3, and 6 h of in vitro germination of olive cultivar Picual pollen were analysed by RNA-seq. A bioinformatics R workflow called *RSeqFlow* was developed contemplating the best practices described in the literature, covering from expression data filtering to differential expression and clustering, to finally propose hub genes. The resulting olive pollen transcriptome consisted of 22,418 reliable transcripts, where 5364 were differentially expressed, out of which 173 have no orthologue in plants and up to 3 of them might be pollen-specific transcription factors. Functional enrichment revealed a deep transcriptional reprogramming in mature olive pollen that is also dependent on protein stability and turnover to allow pollen tube emergence, with many hub genes related to heat shock proteins and F-box-containing proteins. Reprogramming extends to the first 3 h of growth, including processes consistent with studies performed in other plant species, such as global down-regulation of biosynthetic processes, vesicle/organelle trafficking and cytoskeleton remodelling. In the last stages, growth should be maintained from persistent transcripts. Mature pollen is equipped with transcripts to successfully cope with adverse environments, even though the in vitro growth seems to induce several stress responses. Finally, pollen-specific transcription factors were proposed as probable drivers of pollen germination in olive trees, which also shows an overall increased number of pollen-specific gene isoforms relative to other plants.

## 1. Introduction

Flowering plants generate pollen grains that germinate on the stigma and produce a pollen tube (PT)—an evolutionary innovation that is one of the fastest-growing plant-cell-derived structures—to securely deliver sperm cells into the maternal reproductive tissues toward the ovules [1,2]. Pollen tube growth is, therefore, essential for successful fertilisation, becoming of major importance from an agronomical viewpoint. Its development starts once the pollen grain lands on the stigma, where both surfaces are recognised by means of glycosylated proteins. When both surfaces are compatible, reactive oxygen species (ROS) of the stigma are decreased to promote pollen hydration [3] and PT emergence through the pollen grain aperture after 30–45 min [4,5]. The PT grows very rapidly through the style towards the ovules in an energy-demanding process where apoplasmic sucrose accumulated in mature pollen is consumed without any need for sucrose uptake [6]. Only the PT tip enlarges [7], depending on cytoskeleton reorganisation for organelle and vesicle trafficking [8] in a profilin-modulated assembly of actin microfilaments [9] with cell wall biosynthesis at the tip, generation of ROS and nitric oxide, pH homeostasis, energy metabolism, protein folding and degradation (reviewed in [1]), Ca${}^{2+}$ gradient and other ion fluxes, together with aquaporins-dependent water transport, to avoid osmotic stress in coordination with ion transporters in the trans-Golgi network and plasma membrane proton pumps [10]. An increase in autophagy due to the high metabolic demand of PT growth was recently reported [11]. Flavonoids participate in successful pollen germination by increasing the phenylpropanoid pathways [12], ROS scavenging and maintenance of turgor and normal morphology [13,14]. Morphology of PTs depends on ions and water filling the new volume [15,16] and the cell wall that sustains turgor pressure [17,18].

The rapid growth of PT provokes the misfolding of many proteins after their fast synthesis in ribosomes, which requires (i) additional expression of heat shock proteins (HSPs) to stabilise them and, for failing cases, (ii) an increase in some ubiquitination and proteasome components undetectable in mature pollen grains [19]. HSPs are also shielding climate change effects that are exacerbating the intensity, frequency and extremity of heat events, with non-reversible consequences for crops due to the substantial decrease of pollen development, germination, and PT elongation [20]. Plants produce many more small HSPs compared to other eukaryotic organisms, including the HSP20 proteins (presenting a significant homology to eye lens α-crystallins) responsible for heat stress endurance, desiccation tolerance, and lipid stabilisation at the plasma membrane level [21,22]. Interestingly HSP20s accumulate in dry seeds, rapidly disappearing upon germination [22]. F-box proteins (FBPs) belong to one of the families of ubiquitin-ligating enzymes E3, one of the largest and most diverse protein superfamilies found in plants [23]. For example, nearly 700 FBPs have been predicted in *Arabidopsis* [23,24], a pool approximately 10 times larger than in the human genome. They recognise different substrates for ubiquitination, exerting direct control over a wide range of cellular processes [25,26]. Most FBPs are located in pollen, although their function does not seem to be related to it [24] except the pollen-pistil self-incompatibility mechanism to augment genetic variability [27].

Pollen germination requires an overall increase in gene transcription [4,19] that has attracted scarce research, in contrast to the abundant information about pollen development and maturation. Genomics approaches reported that at least 30% of the plant genome could be involved in this process, with a higher proportion of selectively expressed genes than vegetative tissues [28] as a result of its high specialisation. Transcription, ubiquitin-mediated proteolysis (involving up to 54% of the FBP genes), cell cycle progression, and defence/stress response were critical in the transition from mature pollen to the germinated pollen [28,29]. Moreover, PT growth depends on protein synthesis but is relatively independent of transcription [29], which is now explained by the enucleation of the apical region by the callose plug that drives growth progression without de novo transcription using persistent transcripts [30].

Most pollen research has been conducted on model systems, but there is still a place for non-model systems such as olive tree (*Olea europaea* L. subsp. *europaea*), where research on floral biology has been principally horticultural in orientation [31], and relevant and high-throughput technologies have been applied recently [32,33,34,35,36,37]. Olive is an emblematic and dominant crop in the Mediterranean basin whose relevance is related to the high agricultural and nutraceutical value of the fruits (olives), the landscape shaping, and more recently, its use as a source for renewable biomaterials and fuel [38]. In fact, it is the only species within the genus *Olea* that produces edible fruits [39]. Genome sequencing confirmed that the olive tree has a medium-sized genome that ranges from 1.3 to 1.68 Gb, distributed in 23 chromosomes, but with different gene content (compiled in [40,41]). In our laboratory, we developed ReprOlive (the first database of olive reproductive tissues [32]) that was the precursor of our recent OliveAtlas gathering all transcriptomic experiments performed on Picual cultivar published to date [37].

The olive pollen belongs to the big group of angiosperms that undergo mitosis II after anthesis and produces a less differentiated, longer-lived bicellular pollen, whose generative cell divides mitotically in the PT [31,42]. This division represents an important developmental transition, and it is expected to be associated with the removal of a significant part of ribosomes and transcripts, as described for other plants [1]. Messenger RNA breakdown is likely to be initiated by mRNA deadenylation accompanied by effective protein turnover and the accumulation of transcripts associated with the ubiquitin-proteasome pathway (such as FPBs of the E3 ligases mentioned above) [12]. Gene expression changes during PT growth and pollination are governing the fruit set and olive oil production [43], but the only high-throughput approach to the olive pollen was focused on its interaction with stigma [44]. To gain insights into the biological processes involved in pollen germination and PT growth, here we present the first comprehensive study of the in vitro transcriptome dynamics of olive PT growth during four sequential developmental stages using the genome draft of cultivar Picual [45] as a reference. The automated R pipeline *RSeqFlow* was developed, revealing a pollen transcriptome of 22,418 genes, with 5364 differentially expressed genes (DEGs) during the process. New pollen-specific genes were identified, including transcription factors, as well as proofs of cell reprogramming by transcription regulation and protein modifications mediated by HSPs and F-box proteins. Changes in responses to stresses and the final male gamete development at 6 h were also observed.

## 2. Results

### 2.1. Filtered Datasets Revealed the Olive Pollen Transcriptome

RNA from samples in triplicate from germinated pollen at 0, 1, 3, and 6 h of culture were sequenced, providing 38.1–47.7 million paired-reads (samples P3H_2 and P6H_3, respectively) that, once pre-processed, provided a high quantity of useful reads (83.47–95.67%; Section A.1). Each library was then mapped to the 81,484 transcripts of the ‘Picual’ genome (Oleur v0.61) and to the 50,681 transcripts of the *Olea europaea* var. *sylvestris* transcriptome (Oeu v1). Results in Section A.1 show that mapping onto the ‘Picual’ transcriptome is, as expected, better (69.61–76.06%) than to the wild olive.

Mapping counts were then loaded into *RSeqFlow* to (i) filter unexpressed and low-expressed genes, (ii) normalise the reliable genes, and (iii) inspect the replicates. Up to 24,667 transcripts (30%) were never expressed in any pollen sample, 33,708 transcripts (41%) were somewhat expressed in all samples, and the consistent olive pollen transcriptome comprised 22,418 (28%) reliably expressed transcripts (three replicates with CPM>1; their IDs and expression counts are in filteredData-{DATETIME}.tsv (Section A.2). Libraries devoid of noisy transcripts follow a nearly identical gauss-shaped profile, and samples were grouped as expected (Figure 1; compared to raw data (see *Load data → Inspect raw data* of Report.html file in Section A.2) even though samples P0H_3 and P1H_3 appeared mixed in raw data. The normalisation of filtered datasets maintains the same sample grouping (see plots in *Normalisation → Sample similarities → Sample MDS and PCA* and *Raw sample similarities → Grouping raw samples* of Report.html file in Section A.2) as reflected by the decrease of dispersion: BCV (biological coefficient of variation) decreased from 0.1792 in raw data to 0.1475 in normalised data. The MDS plot also revealed the closeness of samples collected at 3 and 6 h of growth. Consequently, filtered, normalised data are appropriate for further gene expression studies.

### 2.2. Differential Gene Expression during Olive Pollen Germination

Heteroscedasticity of the 22,418 useful transcripts comprising the reliable olive pollen transcriptome was then removed in *RSeqFlow* (saved as normHomoscedCPM-{DATETIME}.tsv in Section A.2) before applying a generalised linear model in the seek of differentially expressed genes (DEGs). Although the Bayesian approach with *eBayes* function is the most common method, *RSeqFlow* is designed to use the *treat* function that yields more biologically relevant genes and produces fewer false positives. Table 1 confirmed that *treat* provided 2.63–7.9 fold fewer DEGs than *eBayes* since the latter includes additional 9120 genes (see the Venn diagram of *Differential Expression → Saving DEGs → Per contrasts* in Report.html file in Section A.2). Up to 54.6% of pollen transcripts may be DEGs using *eBayes*, suggesting an oversize issue.

After contrasting all-against-all samples (Table 1), the first notable finding was that the 0→1 h transition involved a remarkable down-regulation, while up-regulation occurred for the remaining transitions. In addition, the closeness of samples P0H_3 and P1H_3 has not introduced any concerning distortion in DEGs, probably due to the results of the *treat* function. The negligible number of DEGs (10, all up-regulated) for the 3→6 h transition is consistent with the MDS plot in Figure 1 and suggests that most gene expression changes occurred within 3 h of culture. Three genes were DEGs in all contrasts only when *eBayes* was used (see ubiquitousDEGs_{DATETIME}.tsv file in Section A.2), corresponding to Oleur061Scf3360g05017.1 (a poly(A) binding protein), Oleur061Scf3133g07005.1 (a cyclin-like protein) and Oleur061Scf0430g00024.1 (a probable aquaporin).

Tentative transcription factors (TFs) were identified within DEGs as candidates for regulators of the studied process. Using the 2741 predicted TFs in 57 families for *Olea europaea* var. *sylvestris* in PlantTFDB v5.0, 12,095 tentative TFs were identified in ‘Picual’ (Section A.3), 608 of them (5%, see Section A.4) distributed in the different contrasts (numbers within parentheses in Table 1). Because *A. thaliana* only contains 1726 TFs in PlantTFDB, it is clear that the olive tree may contain more TFs, even though the number of TFs in cv. Picual could be an overestimation due to the presence of paralogues or alleles as separate transcripts. However, it is worth mentioning that (i) the STAT family was not represented in wild olive, (ii) the putative 608 TFs in DEGs are distributed in 46 TF families (Section A.4), with no members in families BBR-BPC, GeBP, HRT-like, LBD, LFY, RAV, S1Fa-like, SAP, SRS, YABBY and ZF-HD, and (iii) the first TF families in Section A.4 having >5.0% of its members within DEGs can be considered enriched relative to the total number of TFs in cv. Picual.

### 2.3. New PT-Specific DEGs

The TAIR10 annotations of cv. Picual transcriptome v0.61 was extended (Section A.5) so that 93.7% of DEGs were annotated, 91.6% with a functionally characterised orthologue (Section A.5). However, 173 DEGs (Section A.5) were not annotated in any reference database, which makes them candidates for olive genes specifically involved in pollen hydration and PT growth. The possibility of being artefactual transcripts was discarded since (i) 44 of them were already identified as pollen-specific genes in ReprOlive [32] (19 of them with a distant orthologue in UniProt); (ii) 12 were present in other reproductive tissues; and (iii) three of them were likely TFs orthologous to the *Olea europaea* var. *sylvestris* TFs predicted in PlantTFDB v5.0 (Section A.6). Two of these new TFs, Oleur061Scf0040g05012.1 (up-regulated in the 1→6 h transition) and Oleur061Scf2615g01010.1 (up-regulated in the 0→1 h transition), belong to DEGs identified in ReprOlive pollen transcriptome, corresponding to members of the CPP and bHLH families, respectively. The third one, Oleur061Scf1950g00010.1 (down-regulated in the 1→6 h transition), is somewhat homologous to the CO-like family but was not present in the ReprOlive database.

### 2.4. Functional Enrichment during PT Growth

The relevant cellular functions affected in 0→1 h, 1→3 h, 3→6 h and 0→6 h transitions in Table 1 were then analysed using the *A. thaliana* TAIR10 orthologue in Section A.5. To circumvent a possible confounding result caused by the different number of genes between both species, TAIR10 IDs present both in up- and down-regulated DEGs for the same contrast (2 DEGs in the 0→1 h transition, 3 DEGs in the 1→3 h transition and 25 DEGs in the 0→6 h transition) were removed before functional analyses. GO enrichment focused on the Biological Process aspect was represented as a network (Figure 2) where processes are nodes and the edges are supported by shared DEGs pointing to that process. Identification of functional modules (green and orange labels in Figure 2) was easy because mutually overlapping gene sets tend to cluster together.

Because may processes in Figure 2 do not contain TFs, transcriptional and post-translational regulations must be concomitantly involved in PT growth. When TFs were inspected in the same contrasts (Section A.7), NAC, MYB, and, to a lesser extent, ERF, HSF, and bHLH seemed relevant because they are families containing abundant TFs (Section A.4). Over-represented families were, for example, bZIP, C3H, and C2H2.

Deep expression changes were observed in the 0→1 h transition (0→1 h UP panel of Figure 2; already advanced in Table 1). Up-regulated processes are consistent with the activation of ‘oxidative phosphorylation’ and ‘mRNA surveillance’ KEGG pathways (Section A.9) and ‘ribosome synthesis’ and ‘oxidative phosphorylation’ pathways decrease in the 1→6 h transition (BP-OE_p_6H.p_1H_down.tsv file in Section A.8). Down-regulated processes within the first hour of pollen culture, such as ‘PTGS (post-transcriptional gene silencing)’, ‘autophagy’, ‘endocytosis’, ‘cytoskeleton organisation’, ‘cytokinesis’, or ‘male gamete generation’ (0→1 h DOWN panel in Figure 2) will be recovered in later stages, corroborating the deep reprogramming. Other specific processes disappear within the first hour of hydration (‘pollen germination’, ‘mRNA catabolism’, ‘demethylation’ of histones and DNA), together with related KEGG pathways [‘biosynthesis of amino acids and cofactors’, ‘carbon metabolism’ including ‘glycolisis/gluconeogenesis’, ‘endocytosis’, ‘protein processing at endoplasmic reticulum’, and ‘inositol phosphate pathway’ and its signalling system (Section A.9)].

The 1→3 h transition step is a pure PT-growth stage, and the functional analysis only showed up-regulated processes (1→3 h UP panel in Figure 2), including many of the processes previously down-regulated in the 0→1 h transition mentioned above. Others, such as ‘mitotic division’ and ‘cytokinesis’, ‘male gamete generation’, ‘pollen tube growth’, and ‘cell (cycle) regulation’, may be specifically required for PT growth and differentiation of male gametes. The fact that only 10 up-regulated DEGs in the last stage (3→6 h transition; Table 1) are related to ‘male gamete generation’, ‘response to pH’ (that was already activated in the 0→1 h transition period), and the ‘cell cycle transition’ involving phosphorylation and cyclin-dependent kinases (3→6 h UP panel in Figure 2) suggests that further growing will depend only on persistent transcripts. This is confirmed by the up-regulation of ‘RNA degradation’ and ‘mRNA surveillance’ KEGG pathways (also up-regulated in the 0→1 h transition), also indicating that the required set of transcripts was nearly constant after the 3 h of pollen germination. Moreover, the last mitotic division to obtain the reproductive male gametophytes and their transport on microtubules seems to occur in the 1→3 h transition stage of PT growth.

To obtain an overview of the process, the 0→6 h germination period was inspected (0→6 h UP and DOWN panels at the bottom of Figure 2). It was confirmed the up-regulation of the main expected processes of PT growth (‘endocytosis’, ‘endosome to vacuolar transport’, ‘organelle localisation’, ‘cytoskeleton organisation on microtubules’, ‘autophagy’, ‘PTGS’, ’mitotic division’, ‘cytokinesis’, ’male gamete generation’, ‘regulation of growth’, ‘protein glycosylation’, ‘lipid droplet organisation,’ and ‘response to hypoxia’). More interestingly, the negative regulation of gene expression supports the down-regulation of KEGGs pathways related to the biosynthesis of amino acids, cofactors, nucleotide, fatty acids, and glucans, in agreement with down-regulated biological processes in the 0→6 h DOWN panel in Figure 2. The functional enrichment at the 0→6 h transition also showed down-regulation of processes that are no longer essential for pollen tube growth, such as ‘chloroplast relocation’, ‘DNA methylation’, ‘RNA export from nucleus’, ‘nucleotide metabolism’, ‘amino acid metabolism’, ‘fatty acid biosynthesis’, ‘pollen wall assembly’, ‘response to heat,’ and ‘response to ROS’, combined with the increased response to other stresses (hypoxia, salt, light) and the proteasome pathway.

### 2.5. Coordinated Processes Revealed by Co-Expressed Genes

Because genes under the same regulatory control tend to be functionally related, DEGs were submitted to correlation pattern detection in *RSeqFlow*. Because 5364 DEGs outnumber the 12 samples by far, DEGs having little variant expression profiles (CV<0.2) were removed to avoid computational constraints, resulting in 164 DEGs (see the heatmap in Figure 3). Correlation matrices were calculated using three different, well-established clustering algorithms based on a different method to obtain the optimal number of clusters: (i) AHC (agglomerative hierarchical clustering) combined with dynamic cut-off, (ii) *k*-means included the most repeated *k* after calculating 26 metrics, and (iii) MBC (model-based clustering) exploited dynamic clustering (AHC, *k*-Means and MBC plots in Figure 3). Using AHC, three clusters described increased expression and another three a decreased one, while *k*-means and MBC gather up-regulated DEGs in one cluster, down-regulated DEGs in another cluster, and transcripts in the third only slightly increased (Figure 3). Expression profiles of MBC and *k*-means were slightly different because DEGs in each cluster were not exactly the same. Cluster profiles also corroborate that main up-and-down expression changes were consolidated within the first 3 h of growth.

Functional enrichment analysis was performed again for the *k*-means clusters (Figure 4) because this algorithm is usually considered faster and more reliable than the others, and is preferred for large datasets. Similarly, in Figure 2, not all coordinated processes depend on TFs, corroborating that post-translational modifications may also be important to coordinate the processes. Cluster KM-1 gathers already known down-regulated processes (Figure 2) as well as new ones, such as ‘ABA metabolism’ to regulate lipid metabolism, ‘cell fate’ due to changes in the regulatory networks, and ‘cytoskeleton organisation’ related to spindle assembly. KM-2 contains low-expressed DEGs (scaled mean below −0.5 in the ‘*k*-Means’ plot in Figure 3) that slightly increased their levels over the PT growth period and correspond to responses to stresses mentioned above, supported by the KEGG pathway ‘protein processing in ER’ (Section A.10). It is also revealing the slight increase of ribosomes and peptidyl phosphorylation, and that defence to fungus shares DEGs with response to jasmonate and ethylene.

Finally, up-regulated processes (KM-3 in Figure 4) confirm the findings in Figure 2, and also reveals the ‘inhibition of biosynthetic processes/positive regulation of catabolism’, ‘Ca${}^{2+}$ transport’, ‘transcript termination’ (together with the ‘mRNA surveillance’ KEGG pathway), ‘chlorophyll catabolism’, and ‘negative regulation of cell death’. Hence, coordinated processes based on correlated DEGs are consistent with previous knowledge, with special emphasis on response to stresses, catabolism and signalling.

### 2.6. Hub Genes

Because there are correlated genes that can be organised in coordinated processes (Figure 4), some kind of regulation can be inferred, with especial interest on unknown and uncharacterised genes (Appendices Section A.5 and Section A.6). Key regulators should be extracted from hub gene detection implemented in *RSeqFlow* based on networks, where nodes are genes, and the edges are correlations as surrogates of co-expression (see Report.html, section ‘Networking’ at Section A.2), and hub genes are highly connected, central nodes. A total of 36 hub genes were proposed from AHC, 23 from *k*-means and 25 from MBC, resulting in 38 putative, unique hub genes, 20 of them being common for the three clustering algorithms (‘Hub genes’ in Figure 3). Because TFs, kinases, HSPs and FBPs are relevant groups of hub genes (Table 2), transcriptional and post-translational regulations seem to be involved in the control of olive PT growth.

There are four down-regulated families of hub genes: HSPs, FBPs, other genes, and unknown/uncharacterised genes (Table 2 and Figure 5). The most populated one is HSPs with 16 olive genes (Table 2), most of them (13) orthologous to members of the HSP20 family, as well as two HSP90 and one HSP70. The elevated number of olive orthologues to the same HSP20 suggests that the olive genome contains more paralogues of this class of proteins than *A. thaliana*. The expression level of these olive genes falls after 1 h of germination (‘HSPs’ panel in Figure 5), suggesting their involvement in maintaining the viability of mature pollen. Because one of the members (Oleur061Scf0370g03005.1) is orthologue to a heat stress TF, it is tempting to hypothesise that this hub gene may directly or indirectly control the other members of the group.

The next populated group are FBPs (Table 2) with eight orthologues to the same F-box protein SKIP27 (a component of SCF [ASK-cullin-F-box]: an E3 ubiquitin ligase mediating ubiquitination and subsequent proteasomal degradation of target proteins), and one to EDL3, a protein involved in SCF-dependent proteasomal ubiquitin-dependent proteolysis. The parallelism with HSPs is not only evident regarding the prominence of one TAIR10 orthologue but also in the expression profile: FBP expression becomes negligible after 1 h of germination, which limits their influence to the very early stages of pollen germination.

Another tentative down-regulated hub genes (Table 2) are a cytochrome P450 (involved in the metabolism of glucosinolates), an oleosin (necessary to stabilise lipid bodies to prevent their coalescence), and two uncharacterised genes orthologous to the same TAIR10 ID (AT1G27461). Because their expression profiles at 3 h are higher than HSPs and FBPs, their relevance can be extended to the initial steps of pollen germination.

Using the 30 TAIR10 orthologues to down-regulated hub genes in olive (Table 2) as input for STRING-db, the significant functional network ($p<1.0\times 10^{-16}$; ‘Down-regulated hub genes’ networks in Figure 5) indicates that oleosin, cytochrome P450, and the two TAIR10 orthologues to FPBs are not linked to any other process. However, the tentative HSPs are tightly linked to each other, with the HSP90 (AT5G52640) linking the protein folding process with the highly connected ribosome biogenesis network established by AT1G27461, the TAIR10 orthologue to both ‘unknown or uncharacterised’ genes. HSPs were also related to the 18S pre-rRNA processome.

There are only eight up-regulated olive hub genes that reach the maximum after 1 h of germination, while their transcription level is negligible in mature pollen, which strongly relates them to the PT growth. They are classified among ‘transcription factors’ and ‘other up-regulated DEGs’ (last two blocks of Table 2, and Figure 6). Unfortunately, the four TFs are orthologue to uncharacterised TFs in *A. thaliana*. One has a Zn-finger domain, two probably contain NAC domains, and the fourth may be one subunit of the Mediator co-activation complex. Using the TAIR10 orthologues to seed STRING-db (‘TFs’ panel in Figure 6), they seem significantly involved in calcium transport and Ca${}^{2+}$/Na${}^{+}$ exchange ($p<1.83\times 10^{-13}$), and regulation or RNA export from nucleus ($p<2.22\times 10^{-5}$). Additionally, it can be observed that (i) the putative Mediator subunit may be related to auxin response due to its association with MYB70, as well as to pest resistance and senescence or responses to wounding by means of LOX4; (ii) the Zn-finger-containing protein might be involved in GTPase-mediated signal transduction located in endocytotic vesicles; and (iii) the TAIR orthologue to the NAC domain-containing protein Oleur061Scf3317g04032.1 is expressed in response to pathogens and hypoxia based on their GO terms, and the STRING-db also relates it to the TF NPX1 controlled by ABA, with plausible participation in water transport to central vacuole mediated by TIP and PIP proteins (that may also be related to Oleur061Scf0430g00024.1, the DEG corresponding to the aquaporin TIP2-1). Hence, these uncharacterised TFs may be controlling the some processes of pollen germination.

The last four olive genes (‘Other up-regulated DEGs’ panel in Table 2 and Figure 6) did not show an apparent functional relation, and their inspection with STRING-db revealed the same functions as above with significant *p*-value. In particular, the TAIR10 orthologue of Oleur061Scf7673g01036 was uncharacterised, but OliveTreeDB v0.61 describes it as a putative C2 NT-type domain-containing protein in trEMBL, corresponding to PROSITE entry PDOC51840. Oleur061Scf1255g08007.1 may be a vacuolar sodium/calcium exchanger with calcium-binding EF-hand that participates in the maintenance of calcium homeostasis in exchange with Na, as reflected by the separate yellow network in ‘Other up-regulated DEGs’ panel in Figure 6, and is also involved in diurnal rhythm (‘photoperiodism’ in Figure 2 and Figure 4) and salt stress response [46]. Oleur061Scf3503g02024.1 may be a proline-rich receptor-like protein kinase PERK1 that appears associated in the STRING-db network with other proteins involved in ion binding, but not transport and also to cellulose synthase (together with the RING/U-box superfamily protein AT3G10910) and other protein kinases involved in cell growth and pistil-specific extensins (KIPK and AT2G36350).

Finally, Oleur061Scf6158g01023.1 seems to contain an RNA-binding YTH domain like ECT5 or CPSF30 (there are 12 *A. thaliana* proteins with this domain in PANTHER 17.0) that has RNA-binding capacity that decreases the mRNA stability, making it susceptible of degradative processes mediated by NURS complex (InterPro ID IPR039278), in agreement with the STRING-db network in Figure 6. Among the proteins containing this domain, those suppressing meiosis-specific genes during gametogenesis or general silencing as indicated by PFAM ID PF04146 and InterPro ID CD21134 may be related to PT growth. Interestingly, these processes are already seen in Figure 2 and Figure 4.

## 3. Discussion

### 3.1. RSeqFlow Pipeline Is Reliable and Expandable

Bioinformatics analyses, excluding the functional enrichment, were performed with our R pipeline *RSeqFlow*, easily configurable in one single file. This enables the claimed repeatability and reproducibility in published papers [47]. Every result is saved within a folder accompanied by an HTML report containing a detailed explanation of results and the rationale of each analytical step, including relevant bibliography (see the Report.html file in Section A.2). This approach facilitates (i) interpretation, (ii) repeating the analysis as many times were necessary with different parameters and data, and (iii)  retaining all executions for further comparisons. *RSeqFlow* uses the best practices described in the literature [48,49,50,51,52,53,54,55,56,57,58,59,60,61,62,63,64,65,66,67,68,69,70] concerning removal of noisy genes (Figure 1), resilience for unequal library sizes, TMM (trimmed mean of M-values) normalisation for differential expression, CTF (counts adjustment with TMM factors) normalisation for correlation analyses, heteroscedasticity removal, best linear models for less than 10 replicates, use of *treat* instead of *eBayes* (Table 1) to increase the biological relevance, use of several clustering methods with objective criteria for the optimal number of clusters, building networks and communities to extract relevant biological functions and remove spurious connections, avoid memory overflow when calculating networks, and select putative hub transcripts as a surrogate of regulators. This is why the number of DEGs in our study is not comparable to previous works [19,71,72], while the functional interpretation is consistent, which agrees with the guidelines that claim for the interpretation of biological processes instead of genes [73,74]. The tables of TAIR10 orthologues (Section A.5) and TFs (Section A.3) are useful for further functional enrichment analyses (Figure 2 and Figure 4). Authors and co-workers are using *RSeqFlow* for a wide range of applications in the olive tree and other plants, which illustrates its suitability in any field of research where differentially expressed genes are required, and further correlations/co-expressions between them are desired, especially in non-model organisms.

### 3.2. The Olive Pollen Transcriptome Contains Pollen-Specific Genes

The mature pollen plus germinating pollen transcriptome was estimated in 22,418 reliable genes (Section A.2), with a certain level of redundancy as deduced from annotations in Section A.5 and Table 2, in agreement with results reported in [37]. The number of genes identified in this work is very close to the 24,672 identified in *Nicotiana tabacum* PT for similar culture times [71], the 25,062 identified in rice [28], as well as the 21,394 in *Pyrus bretschneideri* [75] or the 19,868 of *Lilium longiflorum* [76], all of them higher than the 13,017 genes of poplar [29] and the 4172 genes described for *A. thaliana* [77]. In jasmine, another member of the Oleaceae family, 178,098 gene products were assembled when pollen-pistil interactions were analysed [72].

The extended annotation of olive pollen transcripts presented here (Section A.5) implied that most DEGs (93.7%) were annotated, the majority (91.6%) with a known function. Within the 5364 DEGs (Table 1), 310 were uncharacterised PT-specific genes, including a subset of 173 DEGs without any orthologue in the databases. However, 56 of them were previously sequenced in the reproductive tissues of cv. Picual [32,34], with up to three DEGs being putative TFs (Section A.6). These findings confirmed the existence of unnoticed pollen-specific genes, as previously reported [19,29,71], which should stimulate the molecular study of pollen germination in more detail [34].

### 3.3. Olive Mature Pollen Is Reprogrammed for PT Growth within the First Hour of Germination

The 0→1 h transition is the only comparison with more down-regulated and less up-regulated DEGs (Table 1) resulting in many enriched processes (Figure 2). This behaviour is consistent with previous findings about a global reduction of transcription after pollen germination [71] due to the deep gene reprogramming necessary for the PT emergence after hydration [4,5]. Pollen reprogramming in vitro to acquire totipotency to produce doubled haploid plants has also been reported [78], where hormonal and epigenetic mechanisms involving DNA and histone demethylation and some histone acetylation, as well as autophagy, cell death and cell wall remodelling, have been observed [78]. Particularly, chromatin reorganisation implies an initial decrease of DNA methylation with a subsequent increase in this activity [79], which is consistent with the decreased ‘macromolecule methylation’ and ‘chromatin organisation’ observed in 0→1 h DOWN and 0→6 h DOWN panels in Figure 2. Autophagy in pollen reprogramming towards embryogenesis [78] is also compatible with ‘autophagy’ processes in 0→1 h DOWN, 1→3 h UP and 0→6 h UP panels in Figure 2. Cell wall remodelling was supported by the increase of ‘glycoprotein biosynthesis’ in the 0→6 h UP and the decrease of ‘pollen wall assembly’ in the 0→6 h DOWN panel in Figure 2. Another proof reinforcing the hypothesis of pollen reprogramming is the decrease of PTGS within the first hour of germination (0→1 h DOWN panel in Figure 2), followed by an increase after 6 h of culture (0→6 h UP panel in Figure 2), which is consistent with previous observations [71] indicating that miRNA regulation is involved in PT growth. Since TFs in Figure 6 are clearly necessary after 1 h of germination, they are good candidates to directly or indirectly participate in the required reprogramming. In conclusion, it is strongly suggested that a deep chromatin reorganisation is required to reprogram olive mature pollen towards the PT emergence and that this reprogramming involves both protein and miRNA regulation of chromatin.

### 3.4. Olive Pollen May Express Specific Isoforms

Focusing on the proposed hub genes in Table 2, many TAIR10 genes have several orthologues in olive pollen, suggesting that, for example, olive tree contains more HSP and FBP proteins than *A. thaliana* [23,24]. The same idea can be extracted from repeated TAIR10 IDs in up- and down-regulated DEGs.

Expression profiles of HSPs and FBPs in Figure 5 clearly suggest their involvement in maintaining the viability of pollen grain before landing on stigma due to their rapid decrease to negligible values after 1 h of culture. The fact that many HSP20 genes appear in this process would extend to pollen the findings described in plant seeds [22], where this protein family provides desiccation tolerance, decreasing their expression upon germination. Regarding hub FBPs, their expression is as expected for many pollen FBPs [24,28] and may be more related to pollen-pistil self-incompatibility [27] or removal of unnecessary proteins after pollen maturation [12] than with pollen protein stability [26]. In conclusion, these results also strengthen the importance of the identification of gene isoforms (alleles, paralogues, alternative splicing, etc.) throughout different developmental stages to discover those specifically involved in olive PT growth.

### 3.5. RNA Surveillance during Olive PT Growth

The olive pollen undergoes the second mitotic division in the PT [42,80] with the elimination of a significant part of ribosomes and transcripts [1], a process that also involves the FPBs of the ubiquitin-proteasome pathway [12], as discussed above. RNA expression appears to be a key process during the first 3 h of PT growth, since ‘mRNA surveillance’—detection and degradation of abnormal mRNAs—as biological process and as KEGG pathway appears upregulated in both 0→1 and 1→3 h transitions (Figure 2) and in KM-3 cluster (Figure 4). In addition, the ‘RNA export from nucleus’ is clearly decreased in the 3→6 h transition, ‘nucleotide metabolism’ is down-regulated in the 0→6 h comparison, and ‘transcription termination’ and ‘RNA degradation’ are accompanying mRNA surveillance in KM-3 (Figure 4). Pollen-specific activities of transcription termination, including 3${}^{\prime }$ processing and polyadenylation (Section A.10), were recently described as up-regulated in PT during growth and guidance in *A. thaliana*, these activities are required for full acquisition of fertilisation competence [81]. Oleur061Scf3360g05017.1 (a polyadenylate-binding protein 7)—a ubiquitous DEG initially down-regulated but then up-regulated during all stages of PT growth involved in the mRNA degradation process of KM-3 (Figure 4)—supports the increasing importance of mRNA termination in olive PT growth. It can be then inferred that after 3 h of PT growth, the pool of olive pollen mRNAs is already formed, and there is no longer de novo transcription onwards [30]. Therefore, the later part of PT growth has to be conducted with persisting transcripts, and can also explain why ribosomes synthesis was initially up-regulated in the 0→1 h transition but appears in the co-expression group KM-2 (Figure 4) with a slight increase (Figure 3). The role of persisting transcripts in later pollen tube growth has been recently demonstrated in *A. thaliana* [30].

In concordance with maintaining PT growth without transcription, it was detected an up-regulation of the ‘negative regulation of cell death’ function (KM-3 in Figure 4). The slow increase of the different pathways to maintain the correct protein folding (KM-2 in Figure 4 and up-regulation of ‘ERAD pathway’ in Figure 2) also confirms that olive PT can growth even though from persistent transcripts, in concordance with previous findings in pear [75].

### 3.6. Olive PT Growth In Vitro Is Subjected to Stress and/or Redox Signalling

Functional networks in Figure 2 and Figure 4 contain many biological processes related to responses to stress and/or redox signalling, which have been already reported in other plants [28,29]. Those processes down-regulated within the first hour of culture (responses to ROS, heat, misfolded proteins and mechanical stimuli) can be hypothesised to be present in the mature pollen to prepare it against adverse environmental conditions, while the up-regulated functions (‘responses to hypoxia’, ‘ERAD proteasome pathway’, and ‘salt, light, and biotic stimuli’), might appear to afford the adverse environment during the in vitro PT growth. In fact, the use of in vitro pollen growth instead of an in vivo approach is a limitation of the present work because it misses the complex dialogue between pollen and pistil occurring from the earliest stages of pollen germination all the way to the signalling that culminates in double fertilization. It is likely that this dialogue triggers profound changes in gene expression in germinating pollen and growing pollen tubes. Therefore, several transcriptional responses shown here, which were detected at 3 and 6 h of growth, could be interpreted as stress responses to the in vitro environment, although they might also be related to the pollen tube getting ready for the required ROS signalling between pollen tube and the pistil. This must be further tested in the olive tree by using in vivo conditions. It is known that flavonoids are important for maintaining PT morphology and ROS modulation [12,14] and are probably involved in these responses to hypoxia and ROS to maintain olive pollen viability, turgor, and shape, as well as contributing to signalling [13,14]. Because ROS response is decreasing, it can explain why the flavin-containing compounds are also globally decreased (0→6 h DOWN panel in Figure 2) as well as the ‘isoprenoid biosynthesis’ in KM-1 (Figure 4) in coordination with the metabolism of other lipids.

It must be mentioned the prevalent presence of ‘protein refolding’ and ‘ER organisation’ processes observed in the 0→1 h transition step required for further ‘ERAD pathway’ for misfolded/unfolded proteins observed at 3 h of germination (0→1 UP and 1→3 UP panels in Figure 2, and in KM-2 in Figure 4) accompanied by proteins of the proteasome pathway (Figure 2). HSPs and FPBs may be guiding the ubiquitination and proteasome pathways that are known to regulate virtually all aspects of plant biology [24], including pollen recognition and rejection [25]. It is probable that the rapid synthesis of proteins for the fast olive PT growth could produce incorrectly folded peptides that may require refolding or even specific turnover, as reported in *A. thaliana* [19]. Taken together, those results suggest that mature, desiccated pollen seems to be equipped with a complete set of transcripts that prepares it against the stressing environment before its landing on the stigma; once the PT starts growing throughout the style (in our case, on the in vitro plate), transcripts related to the protective responses disappear, together with unnecessary biosynthetic processes. In fact, the only protective process that must be retained seems to be related to the cell’s turgor pressure and mechanical obstacles that keep the wall under continuous tension [16], as will be discussed below.

### 3.7. Last Mitotic Division of Generative Cells

Olive pollen undergoes mitosis II after anthesis and produces a less differentiated, longer-lived bicellular pollen whose generative cell divides mitotically in the PT [42,80]. The already discussed decrease of nucleotide, protein and metabolite biosynthesis at 6 h is specifically supported by the 3→6 h transition, where (i) only 10 DEGs were detected (Table 1) since their transcriptional profiles at 3 and 6 h of germination are very similar (Figure 1), and (ii) expression levels of up- and down-regulated clusters in Figure 3 are constants at both culture times. Cyclin-dependent kinases and microgametogenesis are the main processes involved (3→6 h UP panel in Figure 2), and ‘cell fate’ and ‘cytoskeleton organisation’ down-regulation in KM-1 together with ‘negative regulation of cell death’ in KM-3 in Figure 4 suggest that the last mitotic division of microsporogenesis [82] leading to male gamete maturation in olive PTs was probably achieved at 6 h of PT growth and that new transcripts are no longer required, even though the pollen tube is not expected to arrive to the ovary in nature before 24 h of growth [31].

### 3.8. Other Relevant Processes during Olive PT Growth

Pollen hydration is the elicitor of PT emergence [4] and, indirectly, may also drive the well-known biological processes of ribosome assembly for intensive protein synthesis, oxidative phosphorylation to afford energy demands from early stages, and the already discussed mRNA surveillance and endoplasmic reticulum reorganisation (to export proteins and mRNA and control misfolded proteins and unprocessed mRNAs). After 1 h germination, the pollen grain is transcriptionally ready to face PT growth by up-regulation of cytoskeleton reorganisation for anterograde/retrograde transport of vesicles and organelles, vesicular trafficking, Ca${}^{2+}$ and other ion fluxes, distribution of reactive oxygen species, pH control, energy metabolism, and protein folding and degradation [1,4,8,9,12,19,72,75,83], as was also supported results in Figure 2 and Figure 4. This may explain why down-regulation of ‘endocytosis’, ‘secretion’ and ‘vesicle transport’ in the 0→1 h transition is followed by up-regulation of ‘organelle transport’, ‘exocytosis’, ‘endosome to vacuole transport’, ‘reorganisation of cytoskeleton’, probably due to the retrograde and anterograde movement of organelles and vesicles, including the gametes [7], that coordinate exocytosis and endocytosis for a fast pollen tube tip growth [2,8].

Ca${}^{2+}$ was known to be a key messenger for PT growth [1] sensitive to inositol signalling [10], and both processes appeared during PT growth. Olive PT Ca${}^{2+}$ transport was up-regulated (KM-3 cluster in Figure 4) while phosphatidyl inositol biosynthesis is significantly down-regulated in the 0→1 h transition (0→1 h DOWN panel in Figure 2 and the corresponding KEGG pathways in Section A.9). The presence of ‘autophagy’, ‘Ca${}^{2+}$ transport’ and ‘phosphatidylinositol biosynthesis’ in Figure 2 and Figure 4 is consistent with recent findings where autophagy is required for the high metabolic demand of PT growth, an activation that depends on Ca${}^{2+}$ increase and inositol decrease [11]. Ion fluxes and pH control are very active during PT growth as reflects the increase of ‘response to pH’ and response to salt/osmosis processes at 6 h of culture and the decrease of ‘ion homeostasis’ (3→6 h UP and 0→6 h DOWN panels in Figure 2 and KM-3 cluster in Figure 4). This may explain the presence of a putative aquaporin TIP2-1 (Oleur061Scf0430g00024.1, DELTA-TIP) among the ubiquitous genes that can be regulated by cyclin-B1-2 (Oleur061Scf3133g07005.1) which is also a ubiquitous gene. Ion balance would be required in olive pollen to maintain polarity through microfilament skeleton, signalling, and oxidative status of male gametes, as described in other species [10,75], as well as to preserve the cylindrical shape and osmolarity of growing PT [84,85,86], because ions and water are used to fill the new volume and sustain turgor pressure [15].

### 3.9. Putative Regulators

A total of 608 TFs distributed in 46 TF families were DEGs during pollen hydration and PT growth (Section A.4). It seems that 11 TF families are not related to PT growth, while other 22 can be considered over-represented, extending the pollen-specific TF families already reported in rice [28]. Because pollen expresses a high proportion of pollen-specialised genes [28,29], uncharacterised pollen genes (Table 2) were extracted following the ‘guilt-by-association’ approach [61,64,87,88]. Down-regulation of HSPs and FPBs, mainly involved in protein folding, refolding or degradation, can explain why the protein refolding and the responses to stresses in Figure 2 appear down-regulated. It is tempting to propose that this mechanism could be directly or indirectly governed by Oleur061Scf0370g03005.1, a heat-shock TF containing a winged helix-turn-helix DNA-binding domain. Additionally, the uncharacterised genes Oleur061Scf1295g03035.1 and Oleur061Scf8276g00009.1 were involved in the ribosome biogenesis, a process that was up-regulated from the first hour of culture onwards (Figure 2 and Figure 4).

Activated hub genes, mainly those considered TFs (Table 2), may be controlling some relevant processes. Because their functional annotation was transferred by homology, it can be suggested that they were never studied before because their expression might be PT-specific. Transcription factors belonging to bHLH, WRKY and MYB are usually found in pollen [72] since they are very large families of TFs, but our co-expression approach also found TFs from other important families. Particularly, Oleur061Scf9115g00013.1 has no orthologue but is described as a NAC domain-containing protein with low identity in trEMBL, while NACs appear relevant gene up-regulators during PT growth in the present study (Section A.7). Its real participation in PT growth is then unknown, and it is likely a PT-specific TF whose functions and expression would merit future research. Another NAC domain-containing hub gene, Oleur061Scf3317g04032.1, seems related to hypoxia, response to pathogens, and water transport to vacuole probably via the aquaporin Oleur061Scf0430g00024.1 because pollen-specific aquaporins in the vacuole have been described important for pollination and PT-growth [89]. In other words, it is tempting to suggest that both NAC domain-containing genes, together with the Zn-finger-containing protein transcript Oleur061Scf3825g03001.1 are good, uncharacterised candidates to coordinate the reprogramming of olive genes during PT growth.

Oleur061Scf7673g01036.1 contains a C2 NT-type domain typical of proteins involved in the intracellular trafficking of chloroplasts and endosomal vesicles; this suggests a putative implication in vesicle and/or organellar movement during PT growth as already suggested in olive [34].

The Na${}^{+}$/Ca${}^{2+}$ exchanger nature of Oleur061Scf1255g08007.1 can be deduced from the yellow network in ‘Other up-regulated DEGs’ panel in Figure 6 and may be involved in the Ca${}^{2+}$ and ion homeostasis in olive PT, in agreement with *A. thaliana* [46].

Oleur061Scf3503g02024.1 is a putative receptor that may transduce signals to different processes. Our data suggest its relation to ion binding and cellulose biosynthesis, even though this last function was not found to be significantly up- or down-regulated.

Finally, functional annotation of Oleur061Scf6158g01023.1 may suggest its involvement in suppressing meiosis-specific genes during gametogenesis by its ability to bind RNA and send it to NURS-based degradation [90]; this may not only relate this gene with PT growth but also with the second mitotic division of the generative cell.

## 4. Conclusions

The R pipeline *RSeqFlow* contemplating the best practices described in the literature provided relevant transcriptional information about olive pollen germination. It revealed a transcriptome of 22,418 genes that underwent a deep cell reprogramming (entailing DNA reorganisation) from mature pollen to 6 h growth pollen tube, where many HSPs and F-box proteins were key proteins. For example, Oleur061Scf3360g05017.1 (a polyadenylate-binding protein 7) is differentially down-regulated in the 0→1 h transition but differentially up-regulated in the remaining growing stages. Besides the well-known processes involved in pollen germination, protein synthesis and oxidative phosphorylation were from very early stages, and autophagy for the later ones, together with an overall reduction of the biosynthetic metabolism. Mature, desiccated pollen seems to be equipped with protecting transcripts that disappear upon germination. Transcription is very active before the 3 h of pollen germination, but after that, the last mitotic division of microsporogenesis seems to occur together with a transcription arrest, making PT growth dependent on persistent transcripts. Many TFs were identified belonging to 22 different families, with two NAC members (Oleur061Scf9115g00013.1 and Oleur061Scf3317g04032.1) and one Zn-finger protein Oleur061Scf3825g03001.1 having a central role. Aquaporins, proteins with C2 NT-type domain or YTH domain, and Na${}^{+}$/Ca${}^{2+}$ exchangers also appeared as relevant hub genes in this study. Hence, the olive tree displays a wider genetic diversity arising from its variate origin, as well as its domestication, giving rise to different alleles, paralogues, alternative splicing, etc., that merit future characterisation.

## 5. Materials and Methods

### 5.1. In Vitro Pollen Germination, RNA Extraction and Sequencing

Samples from mature rehydrated pollen (0 h growth), as well as pollen germinated at 1, 3, and 6 h of in vitro culture, were prepared in triplicate. The procedures to germinate olive pollen and to extract RNA extract were previously described in [43]. The integrity, quality, and quantity of RNA were assessed using the Agilent Bioanalyser 2100 and RNA Nano 6000 Labchip kit (Agilent Technologies, Palo Alto, CA, USA). RNA was converted to cDNA and sequenced by the High-Throughput Sequencing Unit of the University of Malaga, which yielded 2×75 bp reads in their NextSeq 550, generating a total of 40–50 million reads per replicate. Raw reads were deposited as Bioproject PRJEB59024.

### 5.2. Bioinformatic Procedures

Homology, as well as pre-processing and mapping, was conducted on a SUSE Linux Enterprise Server 11SP2 with Slurm queue system and Infiniband network (54/40 Gbps) consisting of 216 nodes with Intel E5-2670 2.6 GHz cores for a total of 3 456 cores and 8.4 TB of RAM.

#### 5.2.1. Closest Homologue

The closest homologue for transcripts in the olive (cv. Picual) transcriptome v0.61 (https://genomaolivar.dipujaen.es/downloads/Sequences (accessed on 2 July 2023)) against the *Olea europaea* var. *sylvestris* v1 transcriptome (https://ftp.ensemblgenomes.ebi.ac.uk/pub/plants/release-57/fasta/olea_europaea_sylvestris/cdna/Olea_europaea_sylvestris.O_europaea_v1.cdna.all.fa.gz (accessed on 2 July 2023)), and the extension of *Arabidopsis thaliana* TAIR10 orthologues were performed using reciprocal comparison based on the fast and sensitive protein aligner *Diamond BLASTp* v2.0.14 [91] using default parameters and –max-target-seqs 1 in –ultra-sensitive mode to get the best hit.

#### 5.2.2. Pre-Processing and Mapping

Raw reads were analysed with *FastQC* v0.11.9 and *MultiQC* v1.11 [92] to inspect if raw data merit the analysis. They were then pre-processed for low-quality regions, contaminant sequences, adapters, etc., using *SeqTrimBB* v2.1.8 (https://github.com/rafnunser/seqtrimbb (accessed on 2 July 2023); an evolution of *SeqTrim-Next* [93]) applying the standard parameters for Illumina paired reads, a minimal length of 40 bases and a contaminant minimal ratio of 0.65. Clean reads were submitted again to *FastQC/MultiQC* to verify sequence artefact removal. The resulting clean reads from every sample were mapped to the *Olea europaea* cv. Picual transcriptome (cDNA v0.61 with 81,484 transcripts) using *Bowtie* v2.4.4 [94] with default parameters and –no-mixed –no-discordant options to only retain appropriately mapped paired-end reads in the SAM file. *SAMtools* [95] helped in sorting and indexing of SAM files to facilitate read counting per transcript with *sam2counts* (https://github.com/vsbuffalo/sam2counts (accessed on 2 July 2023)). This results in a count matrix where rows are transcripts and columns are samples.

#### 5.2.3. RSeqFlow Pipeline

The previously obtained count matrix is the input for our *RSeqFlow* where configuration, filtering, differential expression, correlation/co-expression and networking analyses are conducted automatically. This workflow can be performed on a supercomputer as well as on laptop computers based on Linux, macOS 12.0 or higher, or even Windows. The complete *RSeqFlow* code can be cloned from GitHub (https://github.com/mgclaros/RSeqFlow (accessed on 2 July 2023)). The results presented here were performed using Bioconductor v 3.16 (BiocManager v 1.30.19) and R 4.2.2 (31 October 2022). It creates the folder RSeqFlow1_results_{DATETIME} (from now on, {DATETIME} is the date and time of the execution; see Section A.2) besides the analysed dataset containing tables of correlations, clusters, normalisation, etc., and a comprehensive HTML report explaining the analysis and plots. Customisation of the configure_wf.R file is compulsory (refer to the GitHub Readme file).

**Pre-processing and differential expression:** Within *RSeqFlow*, read counts from cv. Picual cDNAs v0.61 were converted to CPM (counts per million reads) as described in [58] since only inter-sample comparisons of the same gene will be performed. Those transcripts having CPM>1 (a configurable threshold) in at least three replicates were considered reliable and saved at filteredData-{DATETIME}.tsv file in Section A.2. Reliable transcripts were then normalised using the TMM (Trimmed Mean of M-values) method implemented in BioConductor *edgeR* v3.39.6 [50] library to remove systematic technical effects that occur in the data to ensure that technical bias has minimal impact on the results [96]. TMM was selected due to its established robustness [53,59]. Consistency of filtered and normalised data was confirmed by the decrease of dispersion by means of BCV (biological coefficient of variation) [51]. Normalised counts per gene and sample can be found at TMMnormalisedCounts-{DATETIME}.tsv and TMMnormalisedCPMs-{DATETIME}.tsv files, respectively, in Section A.2. Sample comparisons (0→1, 1→3, 3→6, 0→3, 1→6 and 0→6) were conducted by the highest incubation time relative to the lower incubation time in order to interpret up-regulation as the increase of gene expression over the time. Differentially expressed genes (DEGs) were obtained using a generalised linear model (GLM) adapted to binomial distributions based on *edgeR* v3.39.6 and *limma* v3.52.4 libraries. Since this analysis requires that the standard deviation of transcripts is not dependent on the expression level [56], heteroskedasticity was removed using the *limma::voom* function. The combination of *limma::voom* was reported to provide fewer false positives than other [69] and deals better with outliers and unequal library sizes [54]. Another improvement in *RSeqFlow* is that the fitted model was not adjusted with the classical empirical Bayes moderation of *limma::eBayes*, but the *limma::treat* function [48] that is more rigorous and conservative for few replicates. Its rationale is not based on fold-change thresholds, but using a variably increased fold-change depending on the consistency of the gene expression, resulting in more biologically relevant DEGs [48,56,58]. This is why the minimal fold-change (FC) was set to 1.2 (it is configurable). Concerning the adjusted *p*-value, to avoid the misuse of the statistical significance [63,65], the minimal cutoff was set to 0.1 (it is configurable). All DEGs with *limma::eBayes* and *limma::treat* methods can be found at files AllGenes_allContrast_TREAT_{P-cutoff}_{FC-cutoff}_{DATETIME}.tsv and AllGenes_allContrast_eB_{P-cutoff}_{FC-cutoff}_{DATETIME}.tsv, respectively, in Section A.2. DEGs in each contrast have also been saved in a separate file as DEGs_{CONTRAST}_{method}_{P-cutoff}_{FC-cutoff}_{DATETIME}.tsv in Section A.2. Genes that result in differential expression in all contrast using the *limma::treat* approach (or the *limma::eBayes* approach when no DEG was obtained with *limma::treat*) were saved in the ubiquitousDEGs_{DATETIME}.tsv in Section A.2 for further analyses.

**Clustering:** *RSeqFlow* also includes gene clustering to infer co-expression or co-regulation using R functions of *base, stats, cluster, mclust* and *dendextend* libraries. Expression data (counts) were normalised using the CTF (count adjustment with TMM factors) because it is described as more appropriate for correlation studies [68]. CTF-normalised counts were then standardised (scaled) using the *base::scale* function to reduce spurious variations and expression level differences and obtain comparable expression data. CTF-normalised counts per gene and sample were saved at CTFnormalisedCPMs-{DATETIME}.tsv file in Section A.2. Although only DEGs were considered for clustering, computational constraints due to the oversize of genes versus samples were circumvented by the elimination of stable genes defining a coefficient of variance (CV) threshold of CV>0.2 (configurable). The required dissimilarity matrix based on Euclidean distances was derived from Pearson’s correlation *r* using the *stats:cor* function to obtain distances as 1−r. The matrix was mined using Agglomerative Hierarchical Clustering (AHC), *k*-means and model-based clustering (MBC) [97] using functions in libraries *stats*, *NbClust* v 3.0.1 [55], *dynamicTreeCut* v 1.63.1 [98], and *mclust* v 6.0.0 [60]. The optimal linkage method for genes was calculated using *cluster::agnes* function, the optimal number of clusters was statistically inferred using up to 26 different methods with *NbClust::NbClust, dynamicTreeCut::cutreeDynamic* and *mclust::Mclust* functions, the latter exploiting the dynamic clustering based on the Bayesian information criterion [99]. The minimal number of genes per cluster was set to 10 (it is configurable) to strengthen the functional interpretation. Gene profiles of clusters were obtained using the *stats::aggregate* function on the scaled and CPM expression values for each gene. The assignments of DEGs to AHC, *k*-means, and MBC are saved in ClustersCTF-{DATETIME}.tsv file in Section A.2. As a help for users, the optimal cluster for the three methods was analysed with the *cluster.stats* function of the *fpc* v 2.2.9 library to decide which clustering result is preferable.

**Networks:** The previously-used Pearson’s correlation *r* was also used to construct gene networks that will facilitate the study of biological functions [64,70] because genes under the same regulatory control tend to be functionally related [61] based on a ‘guilt-by-association’ approach [87]. A cutoff score of |r|>0.75 was established so that the coefficient of determination $R^{2}$ is >50% ($r^{2}=0.75^{2}=0.56=R^{2}$) meaning that genes exhibiting such a score would have more than the half of their variance in common, leaving only a 44% [(1−0.56)×100] to randomness [49]. Relevant correlations were then filtered by the adjusted *p*-value to decrease the number of correlations without biological (or causation) sense. Meaningful gene pairs were then used to construct undirected networks as described by Provart et al. [52]. The best correlation of clusters was saved in separate tsv files starting by BestCorrelations_ and the cluster name in Section A.2. Because biological networks have been observed to be highly modular, where tightly-connected genes represent organised communities that are expected to conduct common biological functions [57], and maybe are responsible for a common phenotype, networks were then characterised by the node degree and betweenness, Kleinberg’s centrality score [100], hub score, and other network properties, as well as possible communities (modules), using the corresponding functions from the *igraph* v1.3.5 library. Cluster, network and communities are saved as an R object in file List_of_clusters-{DATETIME}.Rds in Section A.2.

**Candidate hub genes**: Tentative hub genes were proposed from ‘ubiquitous’ DEGs (DEGs in all contrast saved above; in many cases, there are no ubiquitous DEG) and ‘outstanding’ DEGs (DEGs having a Kleinberg’s centrality score > 0.9 and a degree > 10—a note that links with at least other 10 nodes in the network; both values are configurable—) as generally recommended [62]. Candidate hub genes were saved as tsv files starting by OutstandingGenes- and ubiquitousDEGs_ in Section A.2 for further analyses as hub gene candidates.

#### 5.2.4. Functional Interpretations

Functional enrichment analyses of gene lists are crucial to interpret the biological implications of DEGs in high-throughput studies. Since the olive tree genome has been recently published [45], its functional annotation is incomplete, requiring the use of *Arabidopsis thaliana* orthologues instead (Section A.5). The R library *clusterProfiler* v4.6 [101] was used, running on default parameters (in all cases, FDR<0.05 and processes with at least 10 DEGs). Input files for the analysis were any of the tsv files starting by AllGenes_allContrast_ or alternatively List_of_clusters-{DATETIME}.Rds (Section A.2). The final report also includes graphs of the most significant GO terms focusing on biological processes (Section A.8) and KEGG pathways (Section A.9).

Unique TAIR10 orthologues of putative hub genes were subjected to *STRING-db* v 11.5 (https://string-db.org/ (accessed on 2 July 2023)) analysis [67] to know if correlation-linked genes share other functionally related associations. This analysis was also used to determine if any biological processes, locations or pathways were specifically highlighted. Default values were used but asked for up to 20 interactions in the second shell. Functional classification of proteins deduced from genes was also conducted in *PANTHER* 17.0 (http://www.pantherdb.org (accessed on 2 July 2023)) [102]. Putative transcription factors in olive tree were inspected using the TF prediction tool of PlantTFDB v5.0 [103] based on their DNA-binding domain predictions, as well as comparisons to transcription factors for wild olive tree contained in the extended PlantTFDB (http://planttfdb.gao-lab.org/index_ext.php?sp=Oeu (accessed on 2 July 2023)).

## Figures and Tables

**Figure 1 plants-12-02894-f001:**
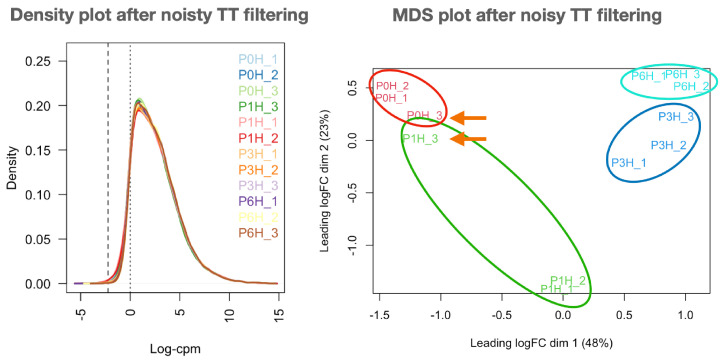
Improved distribution of libraries devoid of noisy transcripts. All libraries follow a similar Gaussian profile (**density plot** at (**left**)) and replicates can be grouped as expected (**MDS plot** at (**right**)). Both plots are provided in Section A.2 and modified for convenience. Sample names in plots are as in Section A.1. Samples from different replicates that are surprisingly close are marked by an arrow.

**Figure 2 plants-12-02894-f002:**
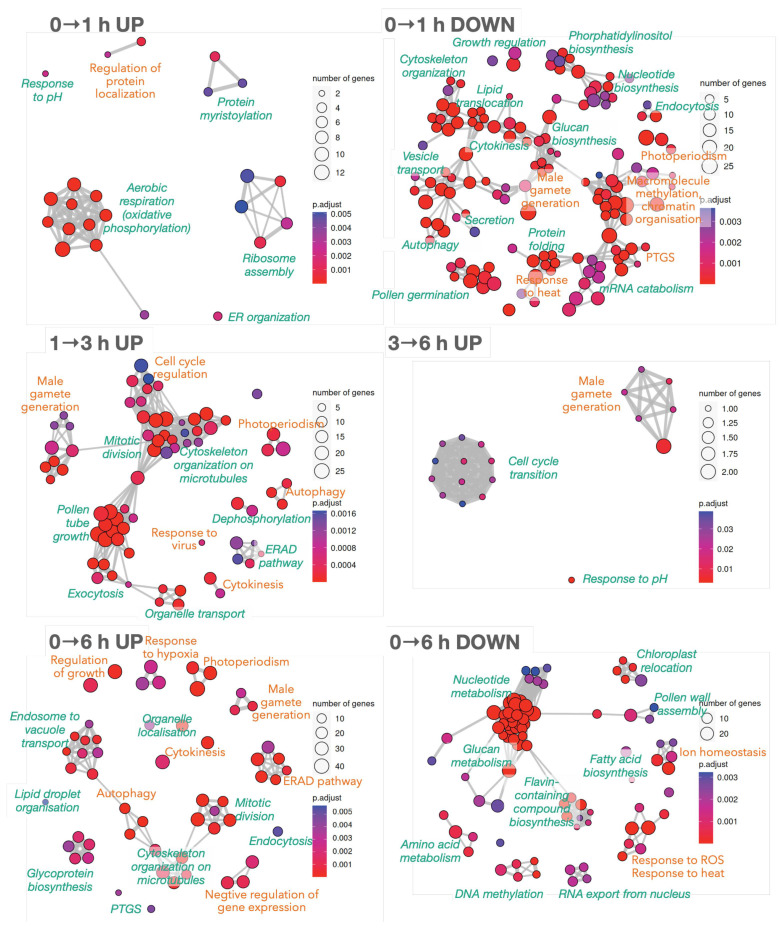
Biological Process enrichment analysis for the DEGs obtained for **0→1 h**, **1→3 h**, **3→6 h** and **0→6 h** transitions of germinating pollen to reveal the main changes from mature pollen to full PT growth in vitro. **UP** refers to processes up-regulated from the first time to the second, and **DOWN** refers to down-regulated processes when advancing from the first time to the second. Green labels close to clusters identify the functional module; labels in orange mark processes containing TFs (Section A.7). Node sizes reflect the number of DEGs supporting the biological process. Blue-to-red colour reflects the significance as an adjusted *p*-value. The complete list of biological processes can be found in Section A.8.

**Figure 3 plants-12-02894-f003:**
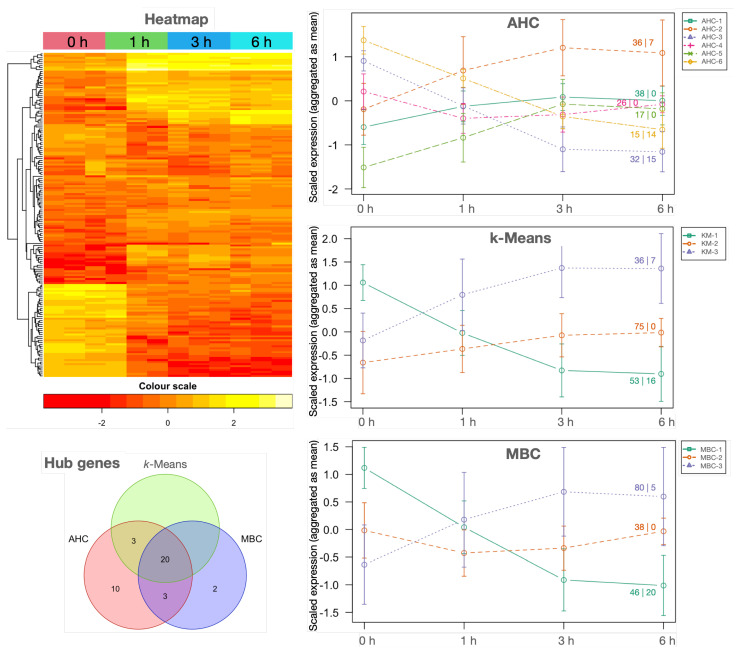
Individual and clustered expression profiles of the 164 highly variable DEGs along PT growth stages. Individual expression in each sample is represented as a **heatmap** (top-left plot). Scaled DEG expression profiles as aggregated medians after clustering using **AHC**, ***k*****-Means** and **MBC** algorithms with standard deviation as error bars are the right plots; the first number on the cluster profile lines corresponds to the number of DEGs aggregated in each point of the cluster and the number after the pipe ‘|’ indicates how many hub DEGs were identified in that cluster. A Venn diagram illustrating the coincidences of the hub DEGs is at the bottom-left (**Hub genes**). The complete list of highly variable DEGs per cluster is available in file ClustersCTF-{DATETIME}.tsv and the putative hubs in each cluster in OutstandingGenes-{DATETIME}.tsv at Section A.2.

**Figure 4 plants-12-02894-f004:**
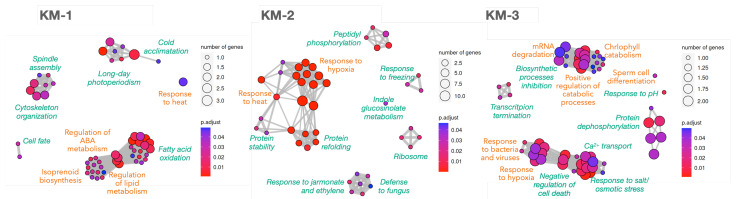
Biological processes enrichment analysis for the 164 highly variable DEGs along PT growth stages after clustering using *k*-means algorithm. The cluster names **KM-1**, **KM-2** and **KM-3** correspond to the same clusters in Figure 3. Green labels close to clusters identify the functional module; labels in orange mark processes containing TFs. Node sizes reflect the number of genes supporting the biological process, as indicated in every legend. Blue-to-red colour reflects the significance as an adjusted *p*-value. The complete list of biological processes and the TAIR10 IDs supporting them can be found in Section A.10.

**Figure 5 plants-12-02894-f005:**
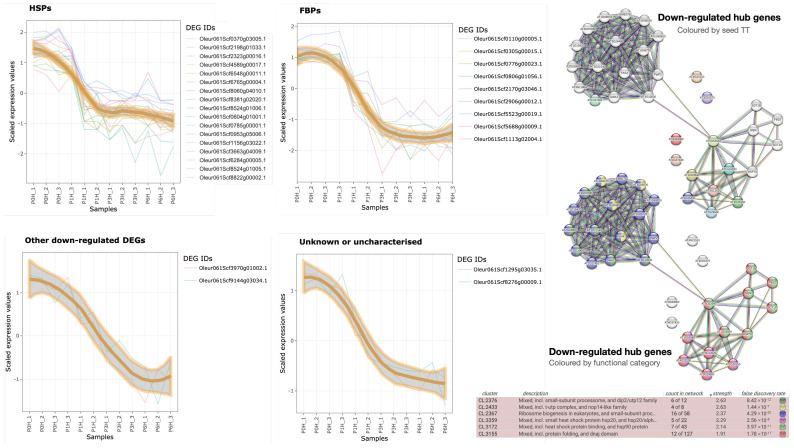
Expression profiles of the four main groups of down-regulated hub genes based on their annotation. The 12 samples were ordered in the abscissa, and the ordinate refers to the scaled transcription level. The functional network of the 30 genes, as revealed by STRING-db, is shown on the right, with input nodes in colours (**top**) or nodes coloured by relevant function (**bottom**).

**Figure 6 plants-12-02894-f006:**
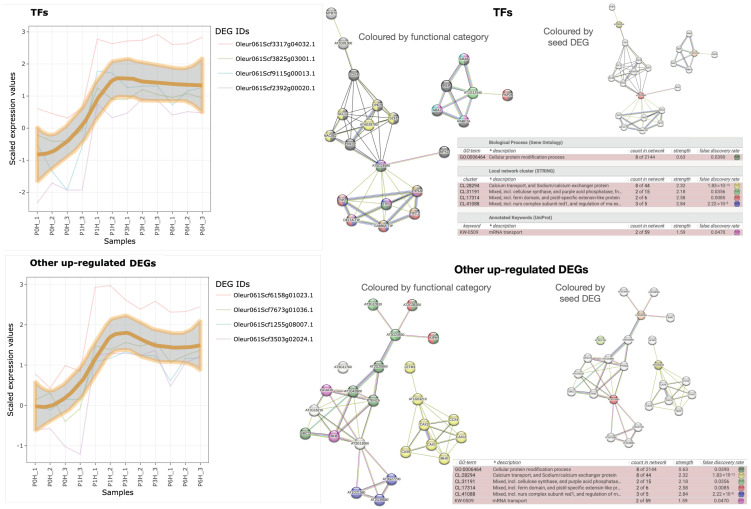
Expression profiles of the main two groups of up-regulated hub genes based on their annotation. The 12 samples are ordered in *x* axis, and the *y* axis refers to the scaled transcription level. Separate functional networks of the 4 TFs and the other up-regulated 4 DEGs, as revealed by STRING-db, are shown on the right, with nodes coloured by relevant functions (big, middle network) or input nodes in colours (small, right network).

**Table 1 plants-12-02894-t001:** Number of DEGs per contrast depending on *eBayes* and *treat* determinations.

DEGs	0→1 h	1→3 h	3→6 h	1→6 h	0→3 h	0→6 h
*eBayes * ${}^{a}$	6715	4776	42	10,348	4015	7087
down	3662	2297	4	4716	1830	3115
up	3053	3479	38	5632	2185	3972
*treat * ${}^{b}$	850 (76)	910 (115)	10 (2)	3929 (429)	948 (133)	2397 (305)
down	561 (49)	119 (13)	0 (0)	1144 (71)	280 (28)	745 (85)
up	289 (27)	791 (102)	10 (2)	2785 (358)	668 (105)	1652 (220)
Ratio *eBayes/treat*	7.90	5.25	4.2	2.63	4.23	2.96

${}^{a}$ All *eBayes* DEGs can be found at file *AllGenes_allContrast_eB_{P-cutoff}_{FC-cutoff}_{DATETIME}.tsv* in Section A.2. ${}^{b}$ The numbers within parentheses are differentially expressed TFs. All *treat* DEGs can be found at file *AllGenes_allContrast_TREAT_{P-cutoff}_{FC-cutoff}_{DATETIME}.tsv* in Section A.2.

**Table 2 plants-12-02894-t002:** Putative hub genes detected after networking analyses of the 164 highly variable DEGs.

‘Picual’ v0.61 ID ${}^{a}$	TAIR10 AC#	Description in OliveTreeDB v0.61
* **Heat shock proteins (HSPs)** * *; down-regulated*		
${}^{1}$ Oleur061Scf2323g00016.1, ${}^{1}$ Oleur061Scf6765g00004.1, ${}^{1}$ Oleur061Scf8060g04010.1, ${}^{1,3}$ Oleur061Scf8524g01006.1, ${}^{1,2}$ Oleur061Scf0785g00001.1, ${}^{1,2,3}$ Oleur061Scf2198g01033.1, ${}^{1,2,3}$ Oleur061Scf0953g05006.1, ${}^{1,2,3}$ Oleur061Scf8524g01005.1	AT1G53540	HSP20-like (class I) chaperone
${}^{1}$ Oleur061Scf8822g00002.1	AT2G29500	HSP20-like (class I) chaperone
${}^{1,2}$ Oleur061Scf3663g04009.1, ${}^{1,2,3}$ Oleur061Scf6548g00011.1	AT5G12020	17.6 kDa class II heat shock protein
${}^{1,2,3}$ Oleur061Scf4589g00017.1, ${}^{1,2,3}$ Oleur061Scf8381g02020.1	AT4G25200	Heat shock 22 kDa protein, mitochondrial
${}^{1,2,3}$ Oleur061Scf1156g03022.1	AT1G16030	Heat shock 70 kDa protein
${}^{1}$ Oleur061Scf6284g00005.1, ${}^{1,3}$ Oleur061Scf0604g01001.1	AT5G52640	Heat shock protein Hsp90 family
${}^{1}$ Oleur061Scf0370g03005.1	AT2G26150	Heat shock transcription factor A2 with winged helix-turn-helix DNA-binding domain
* **Proteins with F-box domain (FBPs)** * *; down-regulated*		
${}^{1}$ Oleur061Scf5523g00019.1, ${}^{3}$ Oleur061Scf1113g02004.1, ${}^{1,3}$ Oleur061Scf0305g00015.1, ${}^{1,2,3}$ Oleur061Scf0776g00023.1, ${}^{1,2,3}$ Oleur061Scf0806g01056.1, ${}^{1,2,3}$ Oleur061Scf2170g03046.1, ${}^{1,2,3}$ Oleur061Scf2906g00012.1, ${}^{1,2,3}$ Oleur061Scf0110g00005.1	AT4G21510	F-box protein SKIP27
${}^{1,2,3}$ Oleur061Scf5688g00009.1	AT3G63060	EID1-like F-box protein 3
* **Other down-regulated DEGs** * *; down-regulated*		
${}^{1}$ Oleur061Scf3970g01002.1	AT4G37430	Cytochrome P450, family 91, subfamily A, polypeptide 2
${}^{1}$ Oleur061Scf9144g03034.1	AT3G01570	Oleosin family protein
* **Unknown or uncharacterised** * *; down-regulated*		
${}^{1}$ Oleur061Scf8276g00009.1, ${}^{1,2,3}$ Oleur061Scf1295g03035.1	AT1G27461	Uncharacterized protein LOC113713235
* **Transcription factors (TFs)** * *; up-regulated*		
${}^{3}$ Oleur061Scf2392g00020.1	AT2G01300	Putative Mediator of RNA polymerase II transcription subunit 18
${}^{1,2,3}$ Oleur061Scf3825g03001.1	AT1G12440	Zinc finger A20/AN1 domain-containing stress-associated protein
${}^{1,2,3}$ Oleur061Scf3317g04032.1	AT5G24590	NAC domain-containing protein 91
${}^{1,2}$ Oleur061Scf9115g00013.1		NAC domain-containing protein
* **Other up-regulated DEGs** * *; up-regulated*		
${}^{1,2}$ Oleur061Scf7673g01036.1	AT3G11760	C2 NT-type domain-containing protein
${}^{1,2}$ Oleur061Scf1255g08007.1	AT1G53210	Sodium/calcium exchanger family protein
${}^{1,2,3}$ Oleur061Scf3503g02024.1	AT3G24550	Proline-rich receptor-like protein kinase PERK1
${}^{1,2,3}$ Oleur061Scf6158g01023.1	AT3G13060	YTH domain-containing family protein 2 putatively regulated by high glucose

^*a*^: Hub genes by AHC (^1^), *k*-means (^2^) and/or MBC (^3^) in OutstandingGenes-{DATETIME}.tsv file in Section A.2.

## Data Availability

RSeqFlow is available at GitHub https://github.com/mgclaros/RSeqFlow (accessed on 2 July 2023). Raw reads are available at Bioproject PRJEB59024. Appendix A data are available in the FigShare project https://figshare.com/projects/Olive_pollen_tube_growth_analysis/171015 (accessed on 2 July 2023).

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
