# Peer review of "Transcriptomic Insight into the Pollen Tube Growth of Olea europaea L. subsp. europaea Reveals Reprogramming and Pollen-Specific Genes Including New Transcription Factors"

_plants, 2023, doi:10.3390/plants12162894_

Round 1

Reviewer 1 Report

Bullones et al. describe the transcriptome of Olea mature pollen and pollen tubes cultured in vitro. The transcriptomes of mature pollen, pollen tubes cultured in vitro, and pollen tubes growing in situ, have all been described before in various species ranging from Arabidoposis to Lilium. However, this study does offer some novel insights, and is notable for meticulous experimental design and rigorous methods deployed for transcriptome analysis.  

Among the original contributions in this manuscript is the careful dissection of the gene expression patterns during hydration and germination versus pollen tube growth from 1 to 6 hours. The authors describe a dramatic reprogramming of transcription during the 0-1 h time interval, as hydration and germination unfold in vitro. The gene expression profile from 1-6 h is markedly different and consistent with cellular processes that are expected to accompany pollen tube growth. That they could pick up a distinctive gene expression profile associated with the division of the generative cell (second pollen mitosis) is impressive. The authors find very little change in the transcript profile during the 3-6 h range, compared to the 1-3 h range, with transcriptional activity essentially ceasing at 6 hours.

One weakness of transcriptome analyses using pollen tubes grown in vitro is that we don’t know how well the observed gene expression patterns apply to pollen germination and tube growth within the pistil.  Other studies have shown a complex dialog between pollen and pistil from the earliest stages of pollen germination all the way to the signaling that culminates in double fertilization, and it is likely that this dialog triggers profound changes in gene expression in germinating pollen and growing pollen tubes. Studies in Arabidopsis have shown, for example, new patterns of gene transcription at later stages of pollen tube growth in the pistil. 

The authors should at least acknowledge the limitation of in vitro transcriptomics in terms of its relevance to how pollen tubes actually grow within the pistil. The time course of pollen tube growth in Olea pistils should be mentioned, so we have context for understanding the in vitro observations and their biological significance. For example, how long does it take for pollen tubes to reach the base of the style on average?  

Certain statements in the manuscript are assumptions on the part of the authors rather than accepted phenomena based on evidence. The notion of ‘culture-induced stresses’ (first mentioned in line 124, page 3) is vague and it does not appear to be factual. The authors should clarify what they mean by ‘culture-induced stresses’. Are they referring to variables unique to in vitro cultures? How do they know that in vitro cultures exert ‘stress’ on the pollen tubes, and what is the nature of this stress?

The article is well-organized and well-written, except for non-idiomatic use of English in a number of instances. For example, in the last sentence of the abstract, ‘gene isoforms relative to other plants’  is a better construction than ‘gene isoforms respect to the other plants’. And in line 45, ‘evolutionary innovations’ is conventional, not ‘evolutive innovation’.

The only issue is the use of some expressions that are not standard in English, even if they 'make sense'. For example, 'lowly expressed' in not conventional in English, and the word 'lowly' has other connotations. The problem can be fixed very simply by writing: low-expressed.

Author Response

We want to acknowledge the referee's comments, since they helped us to improve the current version of our manuscript. Original referee’s issues are in gray.

Bullones et al. describe the transcriptome of Olea mature pollen and pollen tubes cultured in vitro. The transcriptomes of mature pollen, pollen tubes cultured in vitro, and pollen tubes growing in situ, have all been described before in various species ranging from Arabidoposis to Lilium. However, this study does offer some novel insights, and is notable for meticulous experimental design and rigorous methods deployed for transcriptome analysis.  

Among the original contributions in this manuscript is the careful dissection of the gene expression patterns during hydration and germination versus pollen tube growth from 1 to 6 hours. The authors describe a dramatic reprogramming of transcription during the 0-1 h time interval, as hydration and germination unfold in vitro. The gene expression profile from 1-6 h is markedly different and consistent with cellular processes that are expected to accompany pollen tube growth. That they could pick up a distinctive gene expression profile associated with the division of the generative cell (second pollen mitosis) is impressive. The authors find very little change in the transcript profile during the 3-6 h range, compared to the 1-3 h range, with transcriptional activity essentially ceasing at 6 hours.

We sincerely acknowledge the referee for such nice words about our study.

One weakness of transcriptome analyses using pollen tubes grown in vitro is that we don’t know how well the observed gene expression patterns apply to pollen germination and tube growth within the pistil.  Other studies have shown a complex dialog between pollen and pistil from the earliest stages of pollen germination all the way to the signaling that culminates in double fertilization, and it is likely that this dialog triggers profound changes in gene expression in germinating pollen and growing pollen tubes. Studies in Arabidopsis have shown, for example, new patterns of gene transcription at later stages of pollen tube growth in the pistil. 

The referee is absolutely right and we are aware of this weakness, since in our in vitro study pistil was not present and was substituted with a growing medium well known in the field of pollen growth. Although some basic aspects of the interaction between olive pollen and pistil were already studied in our laboratories (Suárez et al, Planta 2013, 237:305–319, DOI 10.1007/s00425-012-1774-z; Castro et al, Ann Bot. 2013, 112(3):503–513, doi: 10.1093/aob/mct118), one may think that single-cell RNA-seq approach could be appropriate for pollen tube growing studies. However, this would require the use of flowers in vivo in their natural orchards and fields (where flowering is asynchronous, there are both hermaphrodite and staminate flowers, the pistil development is delayed in relation to the anthers, and the receptivity of each individual stigma lasts only 2 or 3 days). Natural environments would produce uncontrolled experimental errors, since it would take many years to have olive trees producing flowers in green-houses. To circumvent the limitations of both approaches, there are semi-in vivo methods for other species. Unfortunately, semi-in vivo methods are not appropriate to study the early germination stages, while the in vitro approach enables the investigation of these initial stages but not what would occur when pollen tube gets into the ovary. Moreover, if we were able to adapt the semi-in vivo method to olive flowers, the only way to physically separate pollen tube from pistil is using laser dissection, which is not possible for olive trees. Hence, the in vitro germination of pollen is nowadays the best approach for studies on pollen tube growth, in spite of its limitations.

The authors should at least acknowledge the limitation of in vitro transcriptomics in terms of its relevance to how pollen tubes actually grow within the pistil. The time course of pollen tube growth in Olea pistils should be mentioned, so we have context for understanding the in vitro observations and their biological significance. For example, how long does it take for pollen tubes to reach the base of the style on average?  

Thank you for this comment, which is closely related with the previous one. It is not known the time course of pollen tube in pistils in vivo for olive trees because the typical method of pollen tube cell wall staining using aniline blue is not approachable due to the ovary size. Moreover, only one or two pollen tubes usually penetrate in the ovary locules and arrive to the embrionary sac, resulting in the fertilization of only one or rarely two of the four ovules present in the ovary. The indirect observation of ovary enlarging after fertilisation would be feasible, at this was observed after 48-72 h, as reported by our group in Suaréz et al (2012) Sex. Plant Reprod. 25:133–146, doi 10.1007/s00497-012-0186-3. We deduced then that the tube may had arrived to the ovary after 24-48 h of growth, which is in agreement with results recently published for Camellia oleifera (10.3390/cimb44110366). These ideas were included in the manuscript at the end of section 3.7 in discussion, in lines 548-9:

... male gamete maturation in olive PTs was probably achieved at 6 h of PT growth, and that new transcripts are no longer required, even though the pollen tube is not expected to arrive to the ovary in nature before 24 h of growth [31].

As the referee can see, we were conscious of limitations for the in vitro approach. In fact, we mentioned this issue in the text. The referee’s comment made us think the this idea must be reinforced, and a new sentence was added in section 3.6, lines 504-13:

In fact, the use of in vitro pollen growth instead of an in vivo approach is a limitation of the present work, since it miss the complex dialogue between pollen and pistil occurring from the earliest stages of pollen germination all the way to the signalling that culminates in double fertilization. It is likely that this dialogue triggers profound changes in gene expression in germinating pollen and growing pollen tubes. Therefore, several transcriptional responses shown here, which were detected at 3 and 6 h of growth, could be interpreted as stress responses to the in vitro environment, although they might also be related to the pollen tube getting ready for the required ROS signalling between pollen tube and the pistil. This must be further tested in the olive tree by using in vivo conditions.

Certain statements in the manuscript are assumptions on the part of the authors rather than accepted phenomena based on evidence. The notion of ‘culture-induced stresses’ (first mentioned in line 124, page 3) is vague and it does not appear to be factual. The authors should clarify what they mean by ‘culture-induced stresses’. Are they referring to variables unique to in vitro cultures? How do they know that in vitro cultures exert ‘stress’ on the pollen tubes, and what is the nature of this stress?

We acknowledge the referee for revealing this issue that was not clear in the manuscript. This concept was first used in line 124 of the original document, in Introduction. It is related with the previous issue about the in vitro culture of olive pollen: since it is not growing through the natural location of pistil, and since we detected the induction of some stress responses in Figs 2 and 4, we proposed that some of these stress responses may be induced by culturing on media instead of its natural environment. To clarify this idea, we divided the last sentence of Introduction in two new sentences, one summarising functions for pollen-specific DEGs, and another sentence for the change in responses to stresses and the final male gamete maturation. Lines 123-6 are now as follows:

New pollen-specific genes were identified, including transcription factors, as well as proofs of cell reprogramming by transcription regulation and protein modifications mediated by HSPs and F-Box proteins. Changes in responses to stresses and the final male gamete development at 6 h were also observed.

However, the response to ROS and hypoxia may not be a true stress, but something related to communication between pollen and pistil in the absence of pistil. This prompted us to rename the title of section 3.6 of Discussion as "Olive PT Growth in vitro is Subjected to Stress and/or Redox Signalling" (line 496). Hence, lines 502-4 in this section 3.6 are stating that the up-regulated functions ('responses to hypoxia', 'ERAD proteasome pathway', and 'salt, light, and biotic stimuli'), might appear to afford the adverse environment during the in vitro PT growth was completed with sentences of lines 504-13 mentioned above.

The article is well-organized and well-written, except for non-idiomatic use of English in a number of instances. For example, in the last sentence of the abstract, ‘gene isoforms relative to other plants’  is a better construction than ‘gene isoforms respect to the other plants’. 

Thank you for the recommendation since we are not native English speakers. We changed all "respect to" to "relative to" (lines 40, 184, 712, 904 and 906 of the new manuscript).

And in line 45, ‘evolutionary innovations’ is conventional, not ‘evolutive innovation’.

Thank you for the advice. We changed this word in line 45 of the new manuscript.

Comments on the Quality of English Language:

The only issue is the use of some expressions that are not standard in English, even if they 'make sense'. For example, 'lowly expressed' is not conventional in English, and the word 'lowly' has other connotations. The problem can be fixed very simply by writing: low-expressed.

Thank you very much for this English improvement. We have changed it in lines 136 and 285.

Reviewer 2 Report

1. The section Conclusions is too long and should be shorten. Generally, the description about background and results should not be simply represented or repeated in Conclusions. So I suggest the authors rewrite the sentences concisely.

2. The authors omitted some related closely references to the present research, such as the article "An integrated transcriptomic and proteomic approach to dynamically study the mechanism of pollen-pistil interactions during jasmine crossing". Considering jasmine and Olea europaea L. subsp. Europaea both belong to the Oleaceae family, the suggested reference may be helpful to understand the results.

Author Response

We want to acknowledge the referee's comments, since they helped us to improve the current version of our manuscript. Original referee’s issues are in gray.

1. The section Conclusions is too long and should be shorten. Generally, the description about background and results should not be simply represented or repeated in Conclusions. So I suggest the authors rewrite the sentences concisely.

We acknowledge the referee's suggestion that prompted us to reduce the 37 original lines of Conclusions into only 19 lines (630 to 649), a 51% of the original size. We were not able to produce an even shorter Conclusions since we think that they must contain the most outstanding implications deduced from the results.

2. The authors omitted some related closely references to the present research, such as the article "An integrated transcriptomic and proteomic approach to dynamically study the mechanism of pollen-pistil interactions during jasmine crossing". Considering jasmine and Olea europaea L. subsp. Europaea both belong to the Oleaceae family, the suggested reference may be helpful to understand the results.

We acknowledge this reference and we have discussed it in lines 400 (concerning the number of DEGs), 416-7 (concerning the putative transcriptome genes), 559 (related to the main functions described in pollen tube growth) and 601-3 (concerning transcription factors).